# Self-supervised learning through the eyes of a child

A. Emin Orhan[δ]        Vaibhav V. Gupta[δ]        Brenden M. Lake[δ,ψ]

[δ]Center for Data Science, [ψ]Department of Psychology
New York University
{eo41, vvg239, brenden}@nyu.edu

## Abstract

Within months of birth, children develop meaningful expectations about the world around them. How much of this early knowledge can be explained through generic learning mechanisms applied to sensory data, and how much of it requires more substantive innate inductive biases? Addressing this fundamental question in its full generality is currently infeasible, but we can hope to make real progress in more narrowly defined domains, such as the development of high-level visual categories, thanks to improvements in data collecting technology and recent progress in deep learning. In this paper, our goal is precisely to achieve such progress by utilizing modern self-supervised deep learning methods and a recent longitudinal, egocentric video dataset recorded from the perspective of three young children (Sullivan et al., 2020). Our results demonstrate the emergence of powerful, high-level visual representations from developmentally realistic natural videos using generic self-supervised learning objectives.

## 1   Introduction

Experimental evidence suggests that even very young children have wide-ranging and sophisticated knowledge about the world around them. For example, within the first few months of life, infants show meaningful expectations about objects and agents (Spelke and Kinzler, 2007). Similarly, well before learning to speak, infants can discriminate between many common categories; at 3-4 months, infants can discriminate simple shapes (Bomba and Siqueland, 1983) and animal classes (Quinn et al., 1993), preferring to look at exemplars from a novel class (e.g., bird) after observing exemplars from a different class (dogs). Yet, the origin of this early knowledge is often unclear. How much of this early knowledge can be learned by relatively generic learning architectures receiving sensory data through the eyes of a developing child, and how much of it requires more substantive inductive biases?

This is, of course, a modern reformulation of the age-old nature vs. nurture question that is central in psychology. Answering this question requires both a precise characterization of the sensory data received by humans during development and determining what generic models can learn from this data without assuming strong priors. Although addressing this question in its full generality would require unprecedentedly large and rich datasets and hence still remains out of reach, we can hope to make real progress in more narrowly defined domains, such as the development of visual categories, thanks to new large-scale developmental datasets (Sullivan et al., 2020; Smith and Slone, 2017; Bambach et al., 2018) and the recent progress in deep learning methods.

In this paper, our goal is precisely to achieve such progress by utilizing modern self-supervised deep learning techniques (He et al., 2019; Chen et al., 2020a) and a recent longitudinal egocentric dataset of headcam videos (SAYCam) recorded from the perspective of three developing children (Sullivan et al., 2020). The scale and longitudinal nature of this dataset allows us to train a large-scale model "through the eyes" of individual developing children; in this case, based on ∼150-200 hours of video sampled regularly from 6 months to 32 months of age. Our choice of self-supervised learning avoids extra supervision that a child would not have access to; training only on data from individual children

ensures a strict subset of actual developmental experience. We trained self-supervised models on raw unlabeled videos, with the goal of extracting useful high-level visual representations. The acquired visual representations were then evaluated based on their ability to distinguish common visual categories in the child's environment, using only linear readouts. Our results demonstrate, for the first time, the emergence of powerful, high-level visual representations from natural videos collected from a child's perspective, using generic self-supervised learning methods. More specifically, we show that these emergent visual representations are powerful enough to support (i) high accuracy in non-trivial visual categorization tasks that are behaviorally relevant for a child, (ii) invariance to natural transformations, and (iii) generalization to unseen category exemplars from a handful of training exemplars.

## 2 Related work

In developmental psychology, there is extensive experimental work on the acquisition of perceptual categories in children. As mentioned in the Introduction, 3-4 month old infants can discriminate between many common categories (Bomba and Siqueland, 1983; Quinn et al., 1993), such as dogs vs. birds or triangles vs. squares. It can sometimes be unclear whether infants possess these contrasts before entering the lab—as opposed to acquiring them during the experiment—but other work probes knowledge that is more clearly acquired at home. For example, at 6-9 months, infants already seem to know the meanings of many common nouns referring to food or body-part categories, such as "apple" or "mouth" (Bergelson and Swingley, 2012). Soon after children begin speaking, there is a vocabulary explosion; a six-year-old knows approximately 14000 words, implying they learn about 9 or 10 words a day in early development (Carey and Bartlett, 1978; Bloom, 2002) and suggesting a similarly rapid acquisition of a large amount of categorical knowledge. Although linguistic supervision (e.g. in the form of verbal labels) can guide and sharpen the acquisition of perceptual categories in young infants (Xu et al., 2005), experimental evidence regarding early categorization points to a primarily unsupervised perceptually driven process (Behl-Chadha, 1996; Quinn, 2002).

Learning useful, high-level representations without explicit labels is also a major goal in machine learning. Unsupervised or self-supervised learning methods have been experiencing a robust revival recently, with state-of-the-art self-supervised methods now rivaling the representational power of supervised learning in downstream tasks (He et al., 2019; Chen et al., 2020b,a). Currently, the most successful approaches to self-supervised learning are based on contrastive learning (Hadsell et al., 2006), where the basic idea is to learn nearby embeddings for semantically similar objects (e.g. images or videos) by pushing their embeddings together and distant embeddings for semantically dissimilar objects by pulling their embeddings apart. In the absence of explicit labels to determine semantic similarity, one often uses data augmentation to create semantically similar objects, e.g. by applying color distortions to an image (Chen et al., 2020b). Contrastive self-supervised learning has been applied to both images (Oord et al., 2018; Hjelm et al., 2018; He et al., 2019; Chen et al., 2020b,a) and videos (Sermanet et al., 2018; Zhuang et al., 2020a; Knights et al., 2020) with promising results. However, these works were primarily motivated by computer vision applications, and did not apply self-supervised learning methods to a developmentally realistic, longitudinal, first-person video dataset. Some relatively large first-person video datasets do exist in the computer vision literature, however they are either not longitudinal, e.g. Charades-Ego (Sigurdsson et al., 2018), or they are not developmentally realistic (i.e. not recorded from the perspective of developing children), e.g. KrishnaCam (Singh et al., 2016). Moreover, we are not aware of any prior systematic efforts to apply modern self-supervised learning techniques to such datasets (however, see concurrent work by Zhuang et al. (2020b) training models on the SAYCam data aggregated from multiple children to predict neural responses from the ventral visual system).

Conversely, there are some developmentally realistic, first-person datasets recorded from the perspective of children (Jayaraman et al., 2015; Fausey et al., 2016; Bambach et al., 2018), but these datasets are not longitudinal, instead they were collected from multiple children in relatively short segments. Hence, unlike the SAYCam dataset (Sullivan et al., 2020) that we use in this paper, they are not ideally suited to addressing the fundamental nature vs. nurture question that we are interested in.

Our main contributions in this paper are as follows:

- We show for the first time that it is possible to learn useful, high-level visual representations from longitudinal, naturalistic video data representative of the visual experiences of developing children, using state-of-the-art self-supervised learning methods.

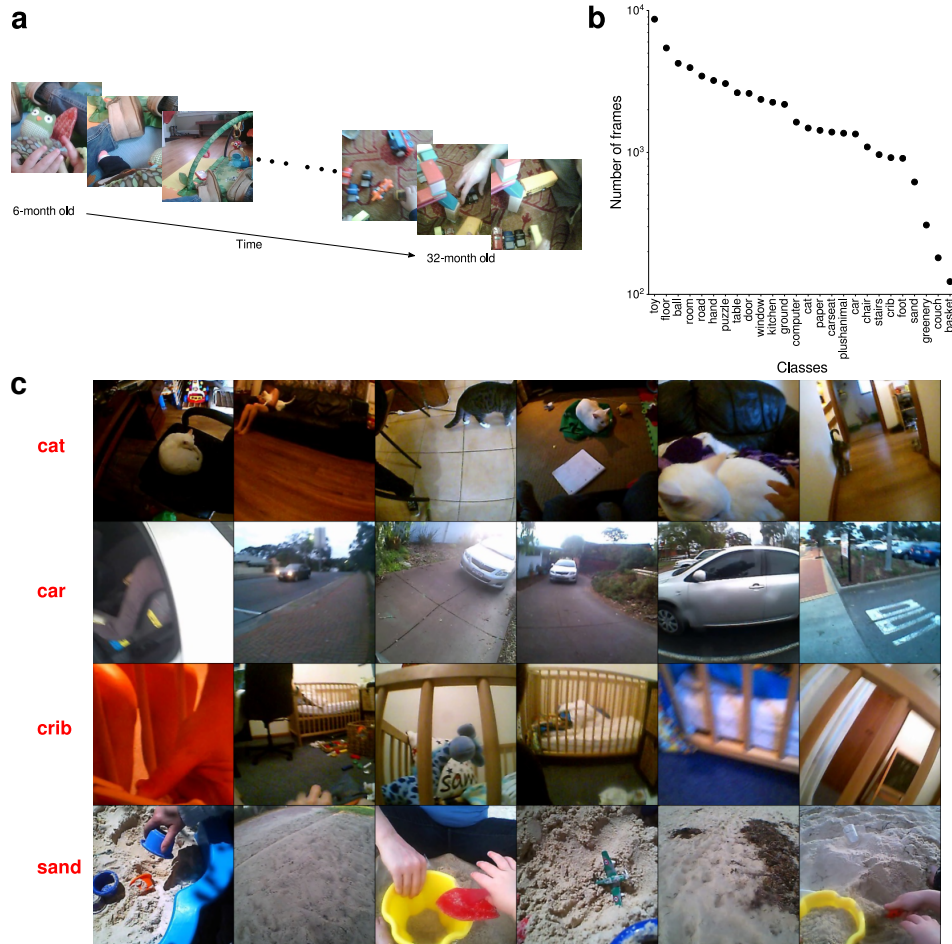

Figure 1: (a) Overall structure of the SAYCam dataset. (b) The classes and the number of frames in each class in the labeled dataset. (c) Random images from the labeled dataset with the category labels indicated on the left. Note that the labeled data are a curated subset of the data from child S.

- We develop a novel self-supervised learning objective for learning high-level visual representations from video data based on the principle of temporal invariance (Földiák, 1991; Wiskott and Sejnowski, 2002) and show that this objective yields better representations than state-of-the-art image-based and temporal contrastive self-supervised learning objectives on the SAYCam dataset.
- From SAYCam, we curate a large, developmentally realistic dataset of labeled images for evaluating self-supervised models and analyze the learned visual representations.

## 3   Dataset

We use the SAYCam dataset (Sullivan et al., 2020) in this study, hosted on the Databrary repository for behavioral science: `https://nyu.databrary.org/`. Researchers can apply for access to the dataset, with approval from their institution's IRB.

This dataset contains approximately 500 hours of longitudinal egocentric audiovisual data from three children: 221 hours from child S, 141 hours from child A, and 137 hours from child Y. The data were collected from head-mounted cameras worn by the children over an approximately two year period (ages 6-32 months) with a frequency of 1-2 hours of recording per week. Figure 1a illustrates the overall structure of the dataset. Although the dataset contains both video and audio data, we only make use of the video component in this paper, as our focus is on studying the development of high-level visual representations. In future work, it would be interesting to consider the potential

benefits of the audio data in the development of high-level visual representations. The native spatial resolution of the videos is 640×480 pixels for all three children and the temporal resolution is 30 frames per second for children A and S, and 25 frames per second for child Y. Before we apply any self-supervised learning algorithms on the data, we first resize the frames (using bicubic interpolation) so that the minor edge is 256 pixels and then take the 224×224 center crop of the frame shifted by 16 pixels upward to exclude the time stamps at the bottom of the video frames that exist for a subset of the videos. A comparative dimensionality analysis of the SAYCam dataset is given in Appendix A1.

**Annotated data.** Although only raw videos are required for training, we need annotated data for model evaluation, preferably from the same dataset. Fortunately, data from one child (child S) comes with rich annotations for ∼25% of the videos, transcribed by human annotators. Using these annotations, we manually curated a large dataset of labeled frames, containing ∼58K frames from 26 classes. Further details on how this labeled dataset was created can be found in Appendix A2. Figure 1b shows the 26 classes in the labeled dataset and the number of frames in each class. Example images from the final labeled dataset are shown in Figure 1c.

# 4   Models

Our modeling effort evaluates the feasibility of learning useful high-level visual representations from a subset of an individual child's visual experience. Our main aim is to measure what is learnable *in principle*, without necessarily constraining the learning algorithms to be strictly psychologically or biologically plausible. With this aim in mind, we trained deep convolutional networks from scratch, using self-supervised learning algorithms on the headcam videos. After training, we evaluated the self-supervised models on downstream classification tasks with developmentally-relevant categories, freezing the trunk of the model and only training linear readouts from the model's penultimate, embedding layer. We used the MobileNetV2 architecture in all experiments below due to its favorable efficiency-accuracy trade-off (Sandler et al., 2018). This architecture has an embedding layer of 1280 units. Pre-trained models and training/testing code are available at: `https://github.com/eminorhan/baby-vision`.

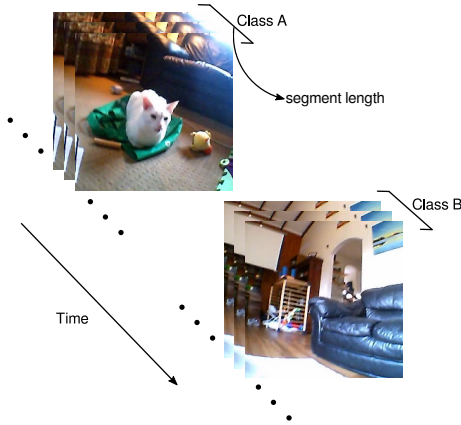

Figure 2: A schematic illustration of the *temporal classification* objective. Each temporal class consists of a number of adjacent frames. The duration of each temporal class is determined by the *segment length* parameter.

**Temporal classification.** To train models on the headcam video data without using any labels, we developed a new self-supervised learning objective based on the principle of *temporal invariance* (Földiák, 1991; Wiskott and Sejnowski, 2002). This objective is based on the observation that higher level variables in a visual scene change on a slower time scale than lower level variables, hence a model that learns to be invariant to changes on fast time scales may learn useful high-level features. We implemented this idea with a standard classification set-up, by dividing the entire video dataset into a finite number of *temporal classes* (or episodes) of equal duration. The objective is then simply to predict which episode any given frame belongs to. A schematic illustration of this *temporal classification* objective is presented in Figure 2. We note that a similar temporal classification algorithm was proposed by Hyvärinen and Morioka (2016) in earlier work as an unsupervised feature learning method to perform non-linear independent component analysis (ICA).

The temporal classification objective relies on psychologically and biologically plausible mechanisms: e.g. it has been suggested that hippocampus provides high-dimensional, sparse, non-overlapping codes for different episodes in an animal's daily life (Marr, 1971). Here, we suggest that such codes can be used as implicit supervision signals for learning useful high-level visual representations.

Importantly, we trained separate models on data from each child to ensure they capture individual rather than aggregate visual experience. Below, we present results demonstrating the effects of various experimental factors on properties of the learned representations (Figure 5): e.g. the frame rate at which the videos are sampled, the segment length parameter (i.e. the duration of each temporal

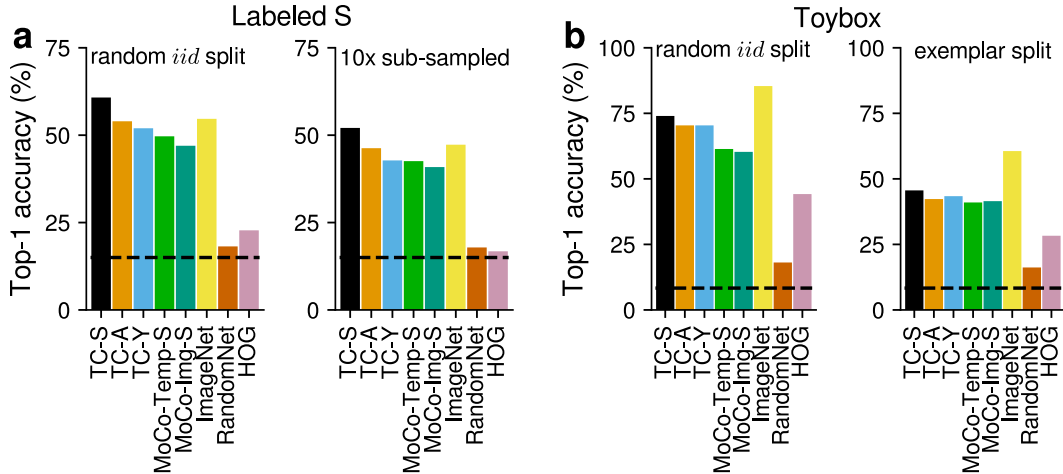

Figure 3: Downstream linear classification tasks: top-1 accuracy of different models in (a) the labeled S dataset, and in (b) the Toybox dataset. Results are shown for both random *iid* splits of the datasets, as well as the more challenging splits discussed in the main text. Horizontal dashed lines indicate the performance of the majority class prediction. Model abbreviations: TC-S, TC-A, TC-Y (temporal classification models for S, A, and Y, respectively), MoCo-Temp-S (a temporal MoCo model trained on data from child S), MoCo-Img-S (a static image-based MoCo model trained on data from child S), ImageNet (an ImageNet-trained model), RandomNet (an untrained, randomly initialized MobileNetV2 model), HOG (histogram of gradients model).

class or episode), and data augmentation. For the remaining results, we always use our best self-supervised model, as measured by classification performance on the downstream classification tasks. Our best model is a temporal classification model that uses a sampling rate of 5 fps (frames per second), a segment length of 288 seconds, and data augmentation in the form of color and grayscale augmentations as in Chen et al. (2020a).

**Static contrastive learning.** To build a strong purely image-based baseline model that does not make use of any temporal information, we trained models with the momentum contrast (MoCo) objective (He et al., 2019) on the headcam data (now treated as a collection of images with no temporal information). We used the "improved" implementation (V2) of MoCo proposed in Chen et al. (2020b). This objective currently achieves near state-of-the-art results on ImageNet among self-supervised learning methods. The basic idea in contrastive learning is to learn similar embeddings for semantically similar ("positive") pairs of frames and dissimilar embeddings for semantically dissimilar ("negative") pairs. We used the PyTorch implementation provided by Chen et al. (2020b) for this model, with the same hyper-parameter choices and data augmentation strategies as in Chen et al. (2020b). Further implementation details can be found in Appendix A3.

**Temporal contrastive learning.** We also trained a temporal contrastive learner that did take the temporal relationship between frames into account. This model is similar to the static contrastive learner above, with the difference that each frame's two immediate neighbors are now treated as positive examples with respect to that frame (temporally non-adjacent frames are still considered as negative pairs as in the static model). Effectively, this model treats temporal jitter between neighboring frames as another type of data augmentation. A similar temporal contrastive learning model was proposed by Knights et al. (2020) before.

**Baseline models.** In addition to the self-supervised models above, we considered several baseline models as controls: (i) an untrained MobileNetV2 model with random weights, (ii) a MobileNetV2 model pre-trained on ImageNet, (iii) HOG features (histogram of oriented gradients) as a shallow baseline (Dalal and Triggs, 2005). For the HOG model, we used the implementation in `skimage.feature`. Further details can be found in Appendix A3.

# 5 Evaluation and analysis of the learned representations

**Downstream linear classification tasks.** As our main evaluation metric, we evaluated the classification accuracy of our self-supervised models, as well as the accuracy of the baseline models, on two downstream linear classification tasks: (i) the curated, labeled subset of the annotated data from child S described above (we call this dataset *labeled S*), and (ii) the Toybox dataset (Wang et al., 2018).

The Toybox dataset is a video dataset consisting of 12 object categories (*airplane*, *ball*, *car*, *cat*, *cup*, *duck*, *giraffe*, *helicopter*, *horse*, *mug*, *spoon*, *truck*), with 30 different exemplars in each category, each undergoing 10 different transformations, such as translations and rotations, for approximately 20 seconds each (plus a brief canonical shot of the object). When the videos are sampled at 1 fps, the entire dataset contains ∼7K frames from each of the 12 categories for a total of ∼84K frames (example images from the dataset are shown in Appendix A4). We chose the Toybox dataset because it contains developmentally realistic objects (toys) belonging to 12 early-learned basic-level categories in child development (Wang et al., 2018). We reasoned that this dataset would thus pose less of a distribution shift problem for our models trained on baby headcam videos, compared to a more complex and diverse dataset such as ImageNet, which is developmentally less realistic (although we do provide results on ImageNet in Appendix A5). Another advantage of the Toybox dataset is that it allows us to evaluate the robustness of self-supervised models to a variety of natural transformations.

Because both the labeled S and the Toybox datasets are video datasets originally, they contain temporal correlations between nearby frames. This raises the potential concern that with random *iid* train-test splits, even models employing relatively low-level strategies might be able to perform well by exploiting these temporal correlations. To address this concern, in addition to evaluating the performance of simple baseline models like the shallow HOG model and the random MobileNetV2 model on these datasets, we also introduced more challenging train-test splits for both datasets. For the labeled S dataset, we reduced the temporal correlations between train and test data by subsampling the entire dataset by a factor of 10 (i.e. an effective frame rate of 0.1 fps). For the Toybox dataset, we introduced an *exemplar split* to examine generalization to novel category exemplars, using 90% of the exemplars for training and 10% for testing (i.e. 27 vs. 3 exemplars from each class). Example images from all evaluation conditions are shown in Appendix A6.

Figure 3 shows the top-1 classification accuracy of all models on the linear classification tasks for both random *iid* splits (with

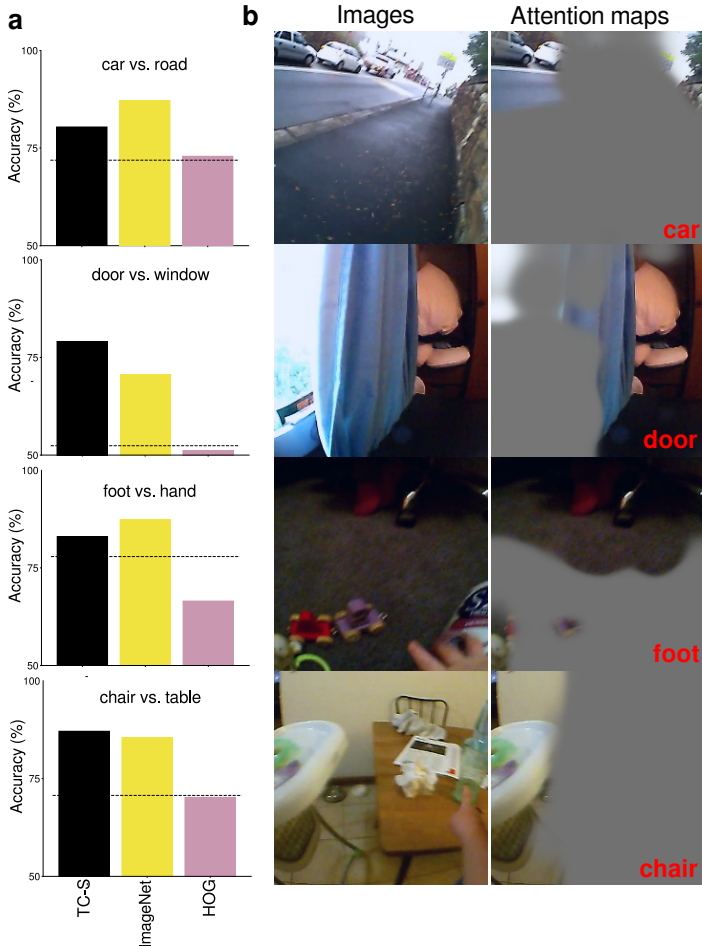

Figure 4: (a) Accuracy of three models on four challenging binary classification tasks. Horizontal lines indicate the majority class prediction accuracy. (b) Example ambiguous images from each task and the spatial attention maps of the self-supervised model (TC-S) for the corresponding images. The model's choice is indicated in red at the bottom right of each picture.

50% training-50% test data) and the more challenging splits discussed above. We observe that the self-supervised models with the temporal classification objective perform well in all cases, sometimes even outperforming the strong ImageNet-trained baseline model. Of particular note is the fact that temporal classification models trained on data from children A and Y are able to generalize well to labeled data from child S. The temporal classification model outperformed the contrastive self-supervised models in all conditions. This may be because the contrastive learning objectives we have considered are not ideally suited to longitudinal video data due to long-range temporal correlations. We leave further investigation of this result to future work. The random MobileNetV2 model and the shallow HOG model generally perform poorly compared to the other models, suggesting that learning and sufficiently deep models are necessary for these tasks. The self-supervised models continue to perform reasonably well on the more challenging splits, suggesting that their performance cannot be explained in terms of simple, low-level strategies.

To give a more intuitive sense of the representational power of the self-supervised models, we also set up four challenging but natural binary classification tasks from the labeled S dataset: *car* vs. *road*, *door* vs. *window*, *foot* vs. *hand*, *chair* vs. *table*. These tasks are challenging because the two classes in each task are semantically related and co-occur in many images, creating ambiguities. Figure 4a shows that the self-supervised TC-S model achieves relatively high accuracy in all tasks (comparable to the ImageNet-trained model) and more importantly the spatial attention maps verify that the model's choices seem to be based on the correct regions in ambiguous images (Figure 4b), suggesting that the model does not just exploit spurious correlations to perform well in these tasks (see below for a description of how these spatial attention maps were computed): e.g. in Figure 4b, note how the model attends to the upper part of the image with the red socks in the third row for a *foot* choice, and the high-chair on the left in the fourth row for a *chair* choice.

**The effect of various experimental factors on downstream linear classification accuracy.** Figure 5 shows the effects of sampling rate, segment length, and data augmentation on classification accuracy in the downstream labeled S task (random *iid* split condition). Higher sampling rates, longer segments (i.e. fewer temporal classes), and using data augmentation all improve classification accuracy in the downstream task. Increasing the sampling rate can be seen as a form of data augmentation, and the effect of these two factors is intuitively clear: data with more variation enables the learning of stronger features (it is, however, interesting to note that the effect of data augmentation on accuracy is rather small; this may be because these egocentric natural videos already come with a lot of natural variability). The dependence of accuracy on segment length, on the other hand, is a function

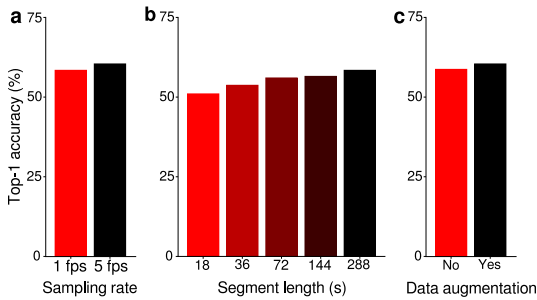

Figure 5: The effects of (a) frame rate, (b) segment length, and (c) data augmentation on classification accuracy in the downstream linear classification task with the labeled S dataset.

of the average time scale over which semantically meaningful changes take place in the videos. It is interesting to note that the optimal time scale appears to be fairly long ($\sim$5 minutes).

**Spatial attention maps.** To better understand what kind of features the self-supervised models rely on to make their decisions in the downstream classification tasks, we computed spatial attention maps from the final spatial layer of the model (`features[18]` or `feats18` for short) using the *class activation mapping* (CAM) method (Zhou et al., 2016). This layer is a $1280\times7\times7$ layer in MobileNetV2 and there is a single global spatial averaging layer between this layer and the output layer. After fitting a linear classifier on top of our best self-supervised model, for each output class in the labeled S dataset, we created a composite attention map from the `feats18` layer by taking a linear sum of all 1280 spatial maps in that layer, where the weights in the linear combination were determined by the output weights for the corresponding class (see Appendix A7 for further details and additional examples).

We then upsampled these composite attention maps and multiplied them with the input image, creating image masks that show where the composite map was most activated. Figure 6 shows examples of masked images for the *cat* class with both actual *cat* images (a) and non-*cat* images (b). These images

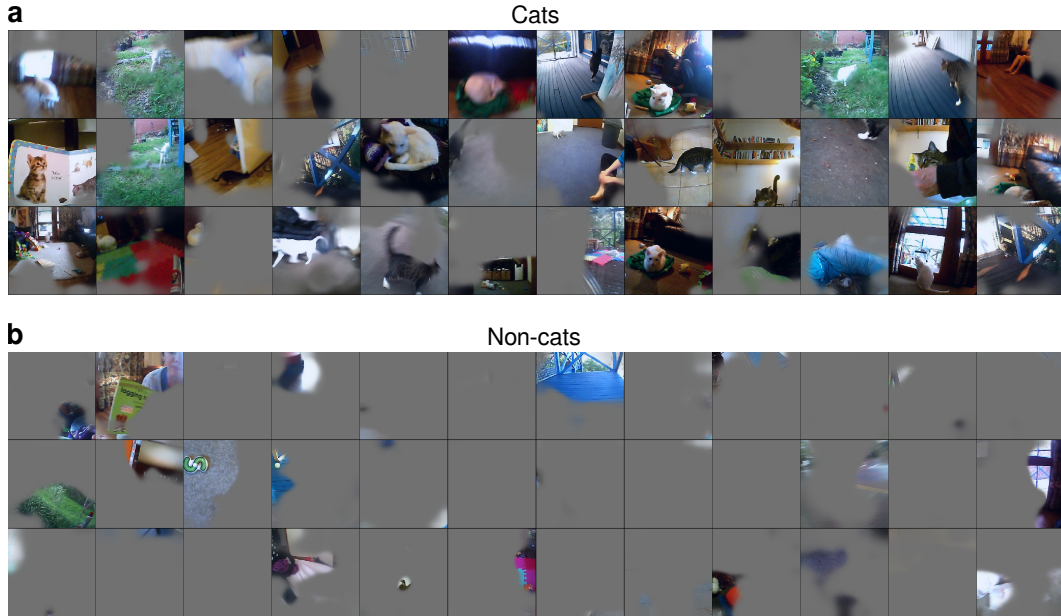

**a** Cats

**b** Non-cats

Figure 6: Masked images indicating the regions in the image the model bases its decisions on. The examples shown are for the output node in the model corresponding to the *cat* class.

suggest that the model, in general, attends to the correct regions in *cat* images, but the spatial extent of attention is usually larger than the cat itself, suggesting possible reliance on contextual cues as well. For the non-*cat* images, the composite map is usually silent, as would be expected from a successful classifier. But there are occasional regions of high activation even in these images, suggesting that the composite maps—and hence the outputs themselves—are not purely class selective.

**Analysis of single feature selectivity.** To measure how distributed vs. localized the representation of class information is, we quantified the class selectivity of individual features, $f$, in the model by the following class selectivity index (Morcos et al., 2018; Leavitt and Morcos, 2020):

$$CSI(f) = \frac{\langle f \rangle_{C_{max}} - \langle f \rangle_{C_{-max}}}{\langle f \rangle_{C_{max}} + \langle f \rangle_{C_{-max}}}$$

where $\langle f \rangle_{C_{max}}$ denotes the average response of the feature to its most activating class and $\langle f \rangle_{C_{-max}}$ denotes its average response to the remaining classes. In computing the average responses, we averaged across the spatial dimensions to obtain a single value per feature per image. The $CSI$ ranges from 0 (for features completely agnostic between classes) to 1 (for features perfectly selective for a single class).

Figure 7a shows the distribution of *CSI*s of individual features in different layers of our best self-supervised model. Single features were generally not very selective for individual classes, pointing to a more distributed representation of class information. The layers close to the output, in general, had higher *CSI*s, consistent with a similar observation made in Leavitt and Morcos (2020) for supervised models. Figure 7b shows 10 highly activating images from the labeled S dataset for 3 example features with high *CSI*s, selective for the *carseat*, *computer*, and *floor* classes, respectively. The example features shown in this figure are from the highest spatial layer of the network (`feats18`). More examples are presented in Appendix A8.

## 6 Limitations

To our knowledge, our work is the first to systematically explore what can be learned from naturalistic visual experience children receive during their development. However, it has several limitations which are important to keep in mind and which would be worthwhile to address in future work.

First, although SAYCam offers an unprecedented look at the experience of individual children, the training videos are a very small fraction of a child's total visual experience, equivalent to ∼1 week

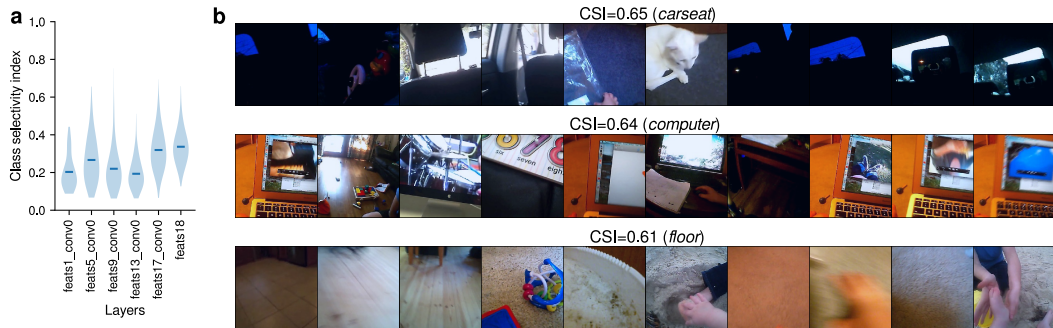

Figure 7: (a) Distribution of *CSI*s in different layers of the model. Horizontal lines indicate the mean of the distribution. (b) 10 highly activating images from the labeled S dataset for three features with high *CSI*s. The most selective classes of these features are indicated in parenthesis. To ensure sufficient diversity in examples displayed here, we first randomly sampled 1024 images from the labeled S dataset, then showed the top 10 most activating images from among this sample.

of visual experience (well-distributed over two years of development). Scaling this up to better approximate a 2.5 year old's experience would require roughly two orders of magnitude more data. Recent results in machine learning suggest that increases in data size on this scale can lead to very large qualitative improvements in model behavior (Halevy et al., 2009; Orhan, 2019; Xie et al., 2019; Brown et al., 2020).

Second, the dataset used in this study includes videos only, hence it ignores the embodied aspect of visual development in children. Humans (and other animals) control their bodies to select the visual experiences they receive. This creates a rich array of sensorimotor inputs that likely help the observer better factorize the sources of variation in their visual experiences. Visual development also has a significant haptic component in animals, especially in dexterous animals like primates, as they can haptically explore the objects around them, which gives them high-quality information about the shapes of objects, for example. Recent computational studies suggest that taking this embodied perspective into account can improve representation learning both in terms of learning speed and in terms of generalization capacity (Jacobs and Xu, 2019; Hill et al., 2019).

Third, and related to the previous point, we also ignored the multimodal nature of cognitive development. Particularly relevant for the development of visual categories is word learning in children. Experimental studies in developmental psychology show that learning object names can change the visual features children use for word learning (Smith et al., 2002; Gershkoff-Stowe and Smith, 2004). We plan to address the role of language through the auditory component of SAYCam in future work.

Fourth, our best self-supervised models currently require unrealistic data augmentation strategies, such as color distortions and grayscaling. It remains to be seen whether equally powerful models can be learned without such unrealistic data augmentation strategies.

## 7   Discussion

In this work, we took a first step toward rigorously addressing a fundamental *nature vs. nurture* question regarding the acquisition of basic visual categories in developing children: can these visual categories be learned through generic learning mechanisms or do they require more substantive inductive biases? By applying modern self-supervised learning algorithms to a strict subset of the visual experiences of individual developing children, we demonstrated the emergence of powerful high-level visual representations, underscoring the power of generic learning mechanisms. Our analysis suggests that although these representations do not strictly correspond to abstract categories (Figure 6), they are abstract enough to support (i) high accuracy in non-trivial downstream categorization tasks (Figure 3), (ii) invariance to natural transformations (random $iid$ conditions in Figure 3) and (iii) generalization to unseen category exemplars (Figure 3b; exemplar split). It still remains open how far we can push generic learning mechanisms, through even larger and richer data sources, to learn mental representations ever closer to those acquired by children early in their development.

## Broader Impact

This research addresses a basic scientific question: *what kind of visual representations can be learned from developmentally realistic, natural video data, using state of the art self-supervised learning methods?* As such, it does not have any significant foreseeable societal consequences and, to the best of our knowledge, the results of this study do not advantage or disadvantage any particular individual or group of individuals. The use of the SAYCam dataset was approved by our Institutional Review Board (IRB) and we followed all applicable guidelines for the use of this dataset.

## Acknowledgements

We are very grateful to the volunteers who contributed recordings to the SAYCam dataset (Sullivan et al., 2020). We thank Jessica Sullivan for her generous assistance with the dataset. This work was partly funded by NSF Award 1922658 NRT-HDR: FUTURE Foundations, Translation, and Responsibility for Data Science.

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
