[Supplementary Material]



Figure A1: Comparative intrinsic dimensionality analysis of four different datasets.

## Appendix

## A1    Comparative dimensionality analysis

Figure A1 shows the variance explained in same sized subsets of 4 different image and video datasets: ImageNet, the headcam videos from child S, and the matched first-person and third-person videos from the Charades-Ego dataset (Sigurdsson et al., 2018). The images or video frames in each dataset were first passed through the largest ResNeXt WSL model with an embedding layer of size 2048 (Mahajan et al., 2018). We then performed a PCA analysis on the embeddings from each dataset, looking at the variance explained as a function of the number of retained dimensions. The video datasets were sampled at 1 fps. Figure A1 shows that, as expected, the video datasets (first-person and third-person videos from the Charades-Ego dataset and the headcam videos from child S) have lower information content than ImageNet, due to temporal correlations in videos. The first-person video datasets (first-person Charades-Ego and the headcam data from child S) have slightly higher information content than the third-person Charades-Ego dataset, presumably because of the higher degree of variability due to natural distortions and perturbations in these first-person videos.

## A2    Curation process for the labeled S dataset

As mentioned in the main text, the headcam data from one of the babies (child S) comes with rich annotations for ~25% of the videos, transcribed by human annotators. Using these annotations, we manually curated a large dataset of labeled frames, containing ~58K frames from 26 classes.The annotations include information such as *the objects being looked at by the child*, *the objects being touched by the child*, and *the objects being referred to*, as well as *the utterances* made, together with approximate time stamps. We used the *the objects being looked at by the child* field to assemble a large collection of labeled frames for evaluation purposes. This field often includes multiple labels for each cell (a cell is the collection of frames between two consecutive time stamps). We only considered the first used label in each cell and performed basic string processing operations to reduce the redundancies in the labels due to annotation inconsistencies (e.g. capitalization, typos, synonymous labels etc.). This reduced the final number of unique labels to 414. We used these labels and the time stamps provided in the annotations to label individual frames in the videos, where we sampled the frames at 1 fps (frames per second).

For evaluation purposes, we further modified this noisy labeled dataset as follows. To obtain a dataset with a sufficiently large number of frames from each class, we restricted ourselves to the top 30 classes containing the largest number of frames. To obtain a balanced dataset, we then removed the top two classes (*mom* and *book*), which contained significantly more frames than the remaining classes. For the remaining classes, to make sure that the labels are clean enough for evaluation, we manually went through each of them, removing frames or changing their labels as necessary. We note that despite our best efforts during this manual cleaning process, some amount of noise and ambiguity might still exist in the labels. We finally removed any classes that contained fewer than

100 frames. This yielded a labeled dataset containing a total of ∼58K frames from 26 classes. The final classes and the number of frames in each class are shown in Figure 1b in the main text.

## A3 Model implementation details

**Self-supervised models.** We trained the temporal classification models with the Adam optimizer with learning rate 0.0005 and a batch size of 732 (maximum batch size we could fit into 4 GPUs using data parallelism). Models trained with 1 fps data were trained for 20 epochs and models trained with 5 fps data were trained for 6 epochs. Final top-1 training accuracy in the temporal classification task was always in the 80-85% range. Before feeding the frames into the model, we always applied the standard ImageNet normalization step (see below). In addition, in the data augmentation conditions, we also applied the probabilistic color jittering and grayscaling transformations from Chen et al. (2020a):

```
transforms.Compose([
transforms.RandomApply([transforms.ColorJitter(0.8, 0.8, 0.8, 0.2)],
p=0.8),
transforms.RandomGrayscale(p=0.2),
transforms.ToTensor(),
transforms.Normalize(mean=[0.485, 0.456, 0.406], std=[0.229, 0.224,
0.225])])
```

For the static and temporal contrastive learning models, we used the PyTorch implementation of MoCo-V2 provided by Chen et al. (2020b) as is, with the same hyper-parameter choices and data augmentation strategies[1]. We trained the models for 6 epochs with headcam video frames sampled at 5 fps. The learning rate was reduced by a factor of 10 in the final epoch.

**HOG model.** For the histogram of oriented gradients (HOG) model, we used the implementation provided in scikits-image (`skimage.feature`) with the following arguments: `orientations=9, pixels_per_cell=(16, 16), cells_per_block=(3, 3), block_norm='L2', visualize=False, transform_sqrt=False, feature_vector=True, multichannel=True`. To fit linear classifiers on top of these features, we used the SGD classifier in scikit-learn (`sklearn.linear_model`), `SGDClassifier`, with the following arguments: `loss='hinge'', penalty=''l2'', alpha=0.0001, max_iter=250`.

## A4 Example images from the Toybox dataset

As mentioned in the main text, we subsampled the videos from the Toybox dataset (Wang et al., 2018) at 1 fps, which resulted in ∼7K images from each of the 12 classes in the dataset. Figure A2 shows example images from each of the 12 classes in the dataset.

## A5 Linear classification results on ImageNet

Although the ImageNet dataset poses a significant distribution shift challenge for models trained on the baby headcam videos, we still evaluated the performance of linear classifiers trained on top of our self-supervised models and obtained the following top-1 accuracies on the ImageNet validation set: TC-S: 20.9%, TC-A: 18.1%, TC-Y: 17.6%, MoCo-V2-S: 16.4%. We also observed that it was possible to achieve close to ∼25% top-1 accuracy with a temporal classification model trained on data from all three children. For comparison, a linear classifier trained on top of a random, untrained MobileNetV2 model (RandomNet) yields a top-1 accuracy of 1.2%. For these ImageNet results, we trained the linear classifiers for 20 epochs with the Adam optimizer using a learning rate of 0.0005 and a batch size of 1024. The training and validation images from ImageNet were subjected to the standard ImageNet pre-processing pipeline.

Figure A2: Example images from the Toybox dataset (Wang et al., 2018). Each row shows 10 random images from a different class. From top to bottom row, the classes are: *airplane*, *ball*, *car*, *cat*, *cup*, *duck*, *giraffe*, *helicopter*, *horse*, *mug*, *spoon*, *truck*.

## A6 Example images from all evaluation conditions

Figure A3 shows example images from the two splits of both datasets used for evaluation in this paper, i.e. labeled S and Toybox.

## A7 Spatial attention maps

The spatial attention maps shown in Figure 4b and Figure 6 in the main text were generated from the final spatial layer of the network, using the *class activation mapping* (CAM) method introduced in

**a**

Labeled S (1 fps)

Labeled S (0.1 fps)

**b**

train       test

random *iid* split

Toybox

exemplar split

Figure A3: (a) Example images from the baseline 1-fps sampling of the labeled S dataset (top) and the $10\times$-subsampled (0.1-fps) version of it (bottom). The shown images are the first 36 frames from the *cat* class in both versions of the dataset. Note that the temporal correlations are substantially reduced in the $10\times$-subsampled version. (b) An illustration of the random *iid* and exemplar splits of the Toybox dataset. The random *iid* split (top row) measures generalization to unseen views (in this case a novel translation of a familiar *airplane*), whereas the exemplar split (bottom row) measures generalization to unseen exemplars (in this case a novel *airplane*).

Zhou et al. (2016). This layer is a $1280\times7\times7$ layer in MobileNetV2. There is a single global spatial averaging layer between this layer and the output layer. After fitting a linear classifier on top of our best self-supervised model, for each output class in the dataset, we created a composite attention map by taking a linear sum of all 1280 $7\times7$ spatial maps, where the weights in the linear combination were determined by the output weights from the corresponding feature to the output node. This results in a single $7\times7$ spatial map for each image. We then upsampled this $7\times7$ map to the image size ($224\times224$) using bicubic interpolation, divided each pixel value by the standard deviation across all pixels, multiplied the entire map by 10 to amplify it and finally passed it through a pixelwise sigmoid non-linearity, i.e. `m<-sigmoid(10.0*m/std(m))`. We then multiplied the attention map with the presented image pixel-by-pixel to obtain the masked images shown in Figure 4b and Figure 6 in the main text. Figure A4 below shows further examples of spatial-attention-multiplied images for the *computer* class.

Figure A4: Example spatial attention maps for the *computer* class in response to (a) computer and (b) non-computer images (similar to Figure 6 in the main text). Note that the attended locations usually make sense for detecting computers.

## A8  Single feature selectivity analysis

Figure A5 below shows further examples of highly activating images for different features in our best self-supervised model (similar to Figure 7b in the main text).

Figure A5: Further examples of highly activating images for different features in our best self-supervised model (similar to Figure 7b in the main text). Each row corresponds to a different feature. These features are all from the final spatial layer of the network (`features[18]`). The *CSI* value of the feature and the most activating class are indicated at the top of each panel. These images were generated in the same way as those shown in Figure 7b in the main text, i.e. to ensure sufficient diversity among displayed examples, we first randomly sampled 1024 images from the labeled S dataset, then displayed the top 10 most activating images from this sub-sample.

## Footnotes

[1]Code available at: `https://github.com/facebookresearch/moco`