[Reviews · NeurIPS 2020]

Review 1

Summary and Contributions: The authors show that training models with self-supervision on a biologically realistic dataset yields useful features for downstream classification tasks.

Strengths: This is a really cool idea. The experiment are clear and the baselines are relatively strong.

Weaknesses: I wish we could have more analysis of when things go wrong. What invariances are missing from the representations. I think a thorough hard example mining could shed some light here. There might even be some insights that some invariances have to be baked in by "nature" (e.g. imagine finding out that all cats seen from the front are misclassified, does that mean that robustness to 3D rotation is not learned?) I hope you'll consider this in future research.

Correctness: Yes

Clarity: Yes

Relation to Prior Work: The paper is framed from a nature vs. nurture perspective, however, I believe the reader would benefit from more context from the computer vision literature. In particular, there are a large number of methods that look at self-supervision from videos. For example: Deep learning from temporal coherence in video, Trading robust representations for sample complexity through self-supervised visual experience, Deep learning of invariant features via simulated fixations in video, Unsupervised learning of spatiotemporally coherent metrics, Unsupervised learning of visual representations using videos, Slow and steady feature analysis: Higher order temporal coherence in video, Transitive invariance for self-supervised visual representation learning. Additionally, there are other surrogate tasks that have been investigated to construct useful representations. For example: Colorization as a proxy task for visual understanding, Unsupervised visual representation learning by context prediction.

Reproducibility: Yes

Additional Feedback: thank you for sharing these cool ideas and results. All the best! ========= After Author Feedback =========== Thank you for taking the time to write a rebuttal. After a brief discussion with the other reviewers and reading the rebuttal I stand by my assessment that this paper should be accepted. All the best.


Review 2

Summary and Contributions: In this manuscript, the authors explore whether meaningful representations for object categorization can be learned through self-supervised training. More importantly, the authors extend the scope of computational modeling to children development by adopting ego-centric video of infants. The authors show that the proposed self-supervised learning method generalizes well to object categorization tasks for two datasets, and the learned representations explain the findings well.

Strengths: Understanding the developmental process of human beings through computational modeling is an important research area, and is of interest to broad audience in the community. The work naturally combines the progress of self-supervised learning algorithm and the SAYCam dataset for infant development recording to help push the boundary of understanding human capabilities for object categorization. To me this is a novel work and is of significance to the research field. Methodologies and experiments are technically sound as well.

Weaknesses: * The authors need to provide more details for the temporal classification method. For example, why and how is the self-supervised approach trained using standard classification setup? Why not training the temporal model using unsupervised ways (for example, combine features across different frames and use single-frame contrastive learning losses, which can be used as a baseline). The authors should provide clear explanations and comparisons for classification setting vs. self-supervised setting. * The authors provide many feature/attention map visualizations. However I am not quite clear about how these single neuron/class level analyses link to the core idea of the manuscript for general object recognition. * Line 193: I am a bit confused about the “exemplar split” setting. Seems using 90% for testing should help with the performance? More explanation is needed. * Figure 3: Why does ImageNet pretrained models perform worse on Labeled S than Toybox? =======Post-rebuttal====== The authors have addressed most of my concerns, including details of temporal classification, unsupervised training baseline, visualizations, etc. I believe the most valuable contribution of the submission is the ego-centric infant video dataset, which helps with the understanding of children development and the corresponding perception capabilities. This definitely enables further research opportunities in the field. I increase the score accordingly.

Correctness: Yes

Clarity: Yes

Relation to Prior Work: Yes

Reproducibility: Yes

Additional Feedback:


Review 3

Summary and Contributions: This paper tackles an interesting question: what can be learned from naturalistic visual experience children receive during their development. Based on a recent longitudinal, egocentric video dataset captured from children, a temporal classification method is proposed, which considering the temporal invariance in model training. The learned model presented competitive transfer learning results on some datasets, compared to MoCo_v2, and even ImageNet supervised pretrained models.

Strengths: + The problem of learning from visual representation of longitudinal, egocentric video dataset is very interesting. I think this problem is quite natural. + The paper is well written. + Systematical experimental exploration of the problem of interest.

Weaknesses: - I expected to see the linear evaluation performance on ImageNet can be impressive. However, it's a pity to see this transfer learning's performance is poor with only TC-S: 20.9% at best. This seriously limits the impact of this work. If the model can only perform well on some easy datasets that are close to the SAYCam, we cannot get too much benefits from learning on such datasets, especially with access to so many big datasets. Maybe the authors can change the SAYCam to other standard videos (Charades) and see if they can have good transfer learning performances. - I expected to see more intuitive discoveries of learning on these egocentric video. But the discovery based on attention is not surprising. The paper uses the method of " Class Activation Maps (CAM)" (Learning Deep Features for Discriminative Localization, CVPR 2016). The recent contrastive learning self-supervised methods also have such properties on just images without considering temporal information. Considering temporal info, there should be more related cues that may be learned, such as object grouping, optical flow, even without using CAM. I suggest authors to read "PsyNet: Self-supervised Approach to Object Localization Using Point Symmetric Transformation", AAAI 2020. - Lack reference: The author uses CAM to visualize attention, but it does not cite the paper. Learning Deep Features for Discriminative Localization, CVPR 2016 - If the transfer learning performances on standard datasets, such as PASCAL, CIFAR, etc can be good. We can say the proposed method can learn good and generalizable performance. Thus may be it's a good idea to report these results. ----Update------ I read the rebuttal and other reviewers' comments and would like to raise the score. Definitely, the problem of learning from longitudinal, egocentric video are quite interesting for both computer vision and psychology community. I may be biased to practical side. But in the future, I hope the authors can show more on what current self-supervised learning (proposed for non-egocentric data) cannot do on egocentric data. And I think all the discoveries in the paper are not surprising and still hold for non-egocentric videos. Thus these discoveries may come from the algorithms side but not from data side. I think the authors do not fully leverage the characters of egocentric videos. If utilized wisely, we may have a better understanding and surpass the current methods proposed for non-egocentric videos/images. But currently as the first attempt, the paper is worthy to be published.

Correctness: Correct.

Clarity: Yes

Relation to Prior Work: Some important missing references.

Reproducibility: Yes

Additional Feedback: Although I think the problem is interesting, I still think this paper can be improved before publications.


Review 4

Summary and Contributions: This paper applies recently proposed unsupervised learning algorithms to a video dataset that is close to what infants receive during their development and shows that self-supervised algorithms can lead to powerful representations that yield strong downstream task performance. --update-- After reading the rebuttal, I will increase my score. I think this work shows a promising new direction for both machine learning and psychology: how to have good machine learning algorithms that can leverage the same data humans get during development. Therefore, I think it’s worthwhile to have the results of the TS algorithm on other Internet datasets. They will show whether different algorithms are needed to handle different datasets.

Strengths: This work is claimed to be the first one showing that a good representation can be learned from a child-view dataset using a self-supervised learning algorithm. The authors also propose a new self-supervised learning algorithm that works better than the SOTA algorithm on this dataset, the temporal classification algorithm. The learning representation from this algorithm even surpasses a ImageNet-pretrained representation evaluated on this dataset. The authors also provide analyses of learned representations.

Weaknesses: Although the learned representation by the TS algorithm is better than ImageNet pretrained representation on the labeled S task, it is much worse than the latter on a third-dataset, Toybox. This indicates that the better performance on the labeled S task is mainly due to the domain advantage this representation has, as it is evaluated on the same dataset that it is learned from. Although the temporal classification algorithm achieves better performance than MoCoV2, the authors could have addressed another interesting question: can this algorithm be applied to other video datasets and learn powerful representations from unlabeled videos that are widely available on Internet? The ablation results show that the segment length of 288s is better than other parameters, however, it would be better to see results of segment length larger than this number.

Correctness: Yes.

Clarity: Yes.

Relation to Prior Work: Yes.

Reproducibility: Yes

Additional Feedback:

[Author Response · NeurIPS 2020]

We thank all four reviewers for their constructive comments. We respond to each reviewer's comments separately below. We also report some results from new experiments suggested by the reviewers.

**R1: (1) More context from computer vision:** We thank the reviewer for suggesting these references, which we will incorporate in an improved discussion of the related work on representation learning from videos.

**R2: (1) More details about temporal classification:** The basic idea in temporal classification is to divide the entire longitudinal video dataset into shorter "episodes" and use these episode labels as implicit training signals: i.e. the model tries to learn which episode a given frame belongs to. This idea is illustrated in Figure 2 in the paper. In the revised paper, we will describe the temporal classification model in more detail. **(2) Training temporal unsupervised models:** First, we emphasize that our temporal classification algorithm is an example of unsupervised (or self-supervised) learning. Second, we followed the reviewer's suggestion and developed a temporal version of MoCo as a new baseline: for each frame, we used the two neighboring frames as "positive" examples with respect to that frame and the remaining frames as "negative" examples. The temporal MoCo model thus tries to learn similar embeddings for neighboring frames. The temporal MoCo model performed slightly better than the purely image-based MoCo model reported in the paper, but still substantially worse than our temporal classification model (e.g. TC-S: 60.4%, MoCo-Temp: 49.3%, MoCo-Img: 46.6% in the labeled S dataset). These new results will be included in the revised paper. **(3) Exemplar split in Toybox:** In Toybox, each of the 12 categories contains 30 exemplars. In the exemplar split, we use 27 of these exemplars to train linear classifiers on top of the pre-trained and frozen features. We use the remaining 3 exemplars for testing. This split is challenging because it requires few-shot generalization, i.e. learning each category from 27 exemplars only. **(4) ImageNet-pretrained models on labeled S vs. Toybox:** ImageNet-pretrained models perform slightly better on Toybox than on labeled S, because it is likely that the Toybox dataset is "more similar" to ImageNet (both are taken by photographers/camerapeople, with a central object, etc.) than the naturalistic headcam videos. **(5) Single feature selectivity analysis:** The main goal of this analysis is to investigate the extent to which the high-level visual categorical information is distributed vs. localized in the self-supervised models. We will do a better job of motivating this important question in the revised paper. The particular analysis used here was inspired by Leavitt & Morcos (2020), who show that distributed representations lead to better object recognition performance. In our case, we have observed a mostly distributed representation of visual categorical information (Figure 7a), giving us assurance that our self-supervised models behave as expected from a high-performing object recognition model.

**R3: (1) Linear evaluation on ImageNet:** ImageNet results reported in the Supplement were obtained without any data augmentation. We have found that it is possible to get slightly better results (22.3% vs. 20.9% top-1 for child S; and 25.2% for a temporal classification model trained on data from all three children) with standard data augmentation methods typically used for this benchmark, i.e. random resized crops and random horizontal flips. Second, there was a typo in the Supplement: top-1 accuracy for a random net should be 1.2%, not 10.2%; so, the self-supervised models perform much better than the random model, suggesting the learned features are indeed quite useful. Third, our goal is not to build a sota model for ImageNet or for any other benchmark. Our main goal in this work is rather to investigate, for the first time, whether it is possible to learn useful, high-level visual representations from developmentally naturalistic videos. SAYCam is much closer to the early visual experience children receive, compared to any other dataset available. So, for our purposes, it doesn't make sense to use some other dataset, as the reviewer suggests, just because it happens to contain more data or higher quality data. If machine learning is to help us understand how the human mind develops, it must account for learning from datasets like SAYCam. **(2) More intuitive discoveries:** In the current work, our focus was explicitly on learning high-level visual categorical representations. In future work, we are also very much interested in exploring the capabilities of the trained self-supervised models further. We thank the reviewer for suggesting object grouping and optical flow as possible properties to investigate further. **(3) Missing references:** We thank the reviewer for pointing out the missing references. After the submission, we have also noticed that the visualization technique we reported in the paper is identical to CAM. We have revised the paper to acknowledge and cite this earlier work. **(4) Transfer learning on further datasets:** We would be happy to include these in the revised paper.

**R4: (1) Application to other video datasets:** The temporal classification algorithm most naturally applies only to longitudinal video datasets and unfortunately we are not aware of very many such longitudinal datasets. Most video datasets in computer vision consist of relatively short video clips instead (typically on the order of tens of seconds); in this setting, the temporal classification model becomes similar to a video instance embedding model, which has been explored before (cf. Zhuang et al., 2020). Also, please note that our main goal in the paper is not to demonstrate the effectiveness of a particular algorithm across a range of datasets, but rather to investigate whether it is possible to learn useful, high-level visual representations from developmentally naturalistic videos (cf. our response **#1** to **R3** above). **(2) Longer segments:** We have indeed tried 576s segments and observed that the model's performance seems to saturate around this point (e.g. 58.6% vs. 58.4% top-1 accuracy in the labeled S dataset for 576s vs. 288s segments, respectively). We expect that the performance will start to deteriorate at some point as the segment length is increased further, but we have not had a chance to try longer segments yet. We would be happy to report new results with longer segments in the revised paper. We thank the reviewer for this suggestion.

[Meta-Review · NeurIPS 2020]

This is an interesting paper combining machine learning and psychology, and brings interesting insights about what can be learned from naturalistic, egocentric, real-world datasets, and how the learned representations can be used on downstream tasks. It’s well-written and clearly presented, and likely of interest to the general NeurIPS audience. Reviewers 3 and 4 initially had concerns about the model’s performance, and suggestions about using other datasets. After discussion and the rebuttal, they were able to be convinced that these run counter to the main motivation of the study. They subsequently raised their scores and all reviewers - and myself - are in agreement to accept.